# Function and Role of ATP-Binding Cassette Transporters as Receptors for 3D-Cry Toxins

**DOI:** 10.3390/toxins11020124

**Published:** 2019-02-19

**Authors:** Ryoichi Sato, Satomi Adegawa, Xiaoyi Li, Shiho Tanaka, Haruka Endo

**Affiliations:** Graduate School of Bio-Applications and Systems Engineering, Tokyo University of Agriculture and Technology, Naka 2-24-16, Koganei, Tokyo 184-8588, Japan; s174092v@st.go.tuat.ac.jp (S.A.); s159217x@st.go.tuat.ac.jp (X.L.); shiho_9099@yahoo.co.jp (S.T.); haruka@edu.k.u-tokyo.ac.jp (H.E.)

**Keywords:** mode of action, 3-domain Cry toxin, functional receptor, ABC transporter, ABCC2, cadherin-like receptor

## Abstract

When ABC transporter family C2 (ABCC2) and ABC transporter family B1 (ABCB1) were heterologously expressed in non-susceptible cultured cells, the cells swelled in response to Cry1A and Cry3 toxins, respectively. Consistent with the notion that 3D-Cry toxins form cation-permeable pores, *Bombyx mori* ABCC2 (BmABCC2) facilitated cation-permeable pore formation by Cry1A when expressed in *Xenopus* oocytes. Furthermore, BmABCC2 had a high binding affinity (*K_D_*) to Cry1Aa of 3.1 × 10^−10^ M. These findings suggest that ABC transporters, including ABCC2 and ABCB1, are functional receptors for 3D-Cry toxins. In addition, the Cry2 toxins most distant from Cry1A toxins on the phylogenetic tree used ABC transporter A2 as a receptor. These data suggest that 3D-Cry toxins use ABC transporters as receptors. In terms of inducing cell swelling, ABCC2 has greater activity than cadherin-like receptor. The pore opening of ABC transporters was hypothesized to be linked to their receptor function, but this was repudiated by experiments using mutants deficient in export activity. The synergistic relationship between ABCC2 and cadherin-like receptor explains their ability to cause resistance in one species of insect.

## 1. Introduction

*Bacillus thuringiensis* is the most widely used bio-pesticide, and it makes several insecticidal proteins including Cry toxins. The largest group of Cry toxins is called 3-domain Cry (3D-Cry) toxin, since its activated form consists of three domains. 3D-Cry toxin genes are essential for the generation of genetically modified insect resistant crops. However, heavy use of 3D-Cry toxin spray formulations and the expansion of cultivation of genetically modified organism (GMO) foods has led to the emergence of insects resistant to 3D-Cry toxins. Determining the causative agent of this resistance is important for developing strategies to prevent resistance development and for understanding the mode of action of 3D-Cry toxins [1,2]. Mapping the locus of the causative gene of a highly resistant tobacco budworm, *Heliothis virescens*, identified ATP-binding cassette (ABC) transporter family C2 (ABCC2) [3]. ABCC2 was also found to be responsible for the Cry1Ac toxin resistance of the diamondback moth, *Plutella xylostella* [4,5]. In Japan, the resistance of several strains of silkworm, *Bombyx mori*, to Cry1Ab was due to insertion of one amino acid residue in ABCC2 [6]. Resistance is thought to be linked to deficiency in factors related to the mode of action of 3D-Cry toxins in the insect midgut. ABCC2 is a membrane-spanning protein consisting of two transmembrane domains (TMD1 and TMD2) and two nucleotide-binding domains (NBD1 and NBD2). The binding affinity of 3D-Cry toxins to brush-border membrane vesicles from resistant strains was lower than that of susceptible strains [7,8]. Therefore, ABCC2 was considered a functional receptor for 3D-Cry toxins, although direct evidence was lacking.

Cadherin-like receptor is also linked to resistance to 3D-Cry toxins [9,10,11,12], albeit a little lower level of resistance than ABCC2 [3,5,13]. The reason for the difference in the levels of resistance was unknown. Why deficiency in each receptor confers resistance in the same insect species was also unclear. For example, YHD2, a Cry1Ac-resistant strain of *H*. *virescens*, harbors several major resistance genes (such as *BtR-4* and *BtR-6*) [10]. It is axiomatic that only deficiency in the most important determinant of susceptibility induces resistance. Typically, deficiency in a less important factor is overridden by the presence of the most important determinant. Therefore, why deficiency in a second receptor also induces resistance in the same species of insect has been unclear.

A *Helicoverpa armigera* strain with 6000-fold increased resistance to Cry2Ab [14] had a mutation in ABC transporter family A2 (*ABCA2*), suggesting that ABCA2 is linked to Cry2Ab resistance [15]. This was confirmed by generating knockout insects using clustered regularly interspaced short palindromic repeats (CRISPR)/Cas9 genome editing [16]. *ABCB1* was found to be linked to 6400-fold increased resistance to Cry3Aa in the leaf beetle, *Chrysomela tremula*, and ABCB1 was identified as a functional receptor for Cry3Aa by heterologous expression in *Spodoptera frugiperda* (Sf9) cells [17]. Cry1 and Cry2 toxins are located on the most distant sides of the 3D-Cry toxin phylogenetic tree, and Cry3 in the middle (Figure 1). Thus, 3D-Cry toxins might have evolved through adaptation from one family to the other families of ABC transporters. ABC transporters export waste substances from cells opening the gate of their latent pore [18]. In addition, 3D-Cry toxin was indicated to insert helices into the cell membrane to make pore [19,20,21,22]. Thus, the working hypothesis was that 3D-Cry toxins insert helices into the opening gate of the pore [23]. Actually, *B. mori* ABCC2 (BmABCC2) exports the fluorescent Ca^2+^ indicator, X-Rhod-1, from Sf9 cells [24]. However, it is unlikely that all 3D-Cry toxins use only ABC transporters as receptors; indeed, Cry1 toxins can also use cadherin-like receptor [9,12], which lacks an intramolecular pore. Therefore, we were curious about the link between opening of the intramolecular pore and the receptor function of ABC transporters. 

Several reviews of the mode of action of 3D-Cry toxins have been published; however, most focused on the role of cadherin-like receptor, aminopeptidase N, and alkaline phosphatase in pore formation [25]. As discussed above, high level resistance is linked to deficiency in ABC transporters, which generated further questions on the mode of action of 3D-Cry toxins.

Here, we review biochemical, cell biological, and physical studies of the role of ABC transporters as receptors for 3D-Cry toxins.

## 2. ABC Transporters are One of the Primary Receptors for Several 3D-Cry Toxins

### 2.1. ABC Transporters Facilitate Induction of Cell Swelling by 3D-Cry Toxins 

3D-Cry toxins cause insect death by inducing cell swelling in the midgut [26]. Incubation of *B. mori* midgut with 100 nM Cry1Aa resulted in swelling of columnar cells, followed by their protrusion from the epithelium and rupture within 60 min. Thus, if ABC transporters are receptors for 3D-Cry toxins, they should induce swelling in non-susceptible cultured cells in the presence of 3D-Cry toxins. At concentrations insufficient to induce cell swelling, 3D-Cry toxins induced apoptosis of columnar cells in *B. mori*, as indicated by nuclear fragmentation, within 2 to 3 days [27]. This programmed cell death promoted repair of damaged midgut tissue by removing injured cells. Furthermore, oncosis-like programmed cell death occurred in S5 cells heterologously expressing cadherin-like receptor [28,29]. However, this oncosis-like programmed cell death pathway has not been reported to induce cell swelling and collapse of the midgut. 

BmABCC2 facilitated induction of swelling by Cry1Aa, Cry1Ab, and Cry1Ac in non-susceptible Sf9 cells when heterologously expressed using a baculovirus expression system [30]. The cells started swelling within 3 min, became non-refractive, reached the maximum volume after 15 min, and burst when they were administrated with 1.2 μM Cry1Aa toxin. As more than 90% of the cells expressed BmABCC2, a quantitative evaluation was conducted [30] (Appendix A). BmABCC2 also facilitated induction of swelling by Cry1As in human HEK293T cells [31]. Sf9 and HEK293T cells are phylogenetically dissimilar and do not have any adaptor molecules for BmABCC2. Therefore, it is not easy to imagine that BmABCC2 triggered the same kind of programed cell death in these two cell lines, although cadherin-like receptor triggered programed cell death and facilitated induction of swelling simultaneously in S5 cells [28,29]. When BmABCC2 was expressed in wing-disc cells of *Drosophila melanogaster* using the Gal4-UAS system, the cells became susceptible, swelled, and died in response to administration of 200 nM Cry1Aa in vitro [32]. In addition, when transgenic *D. melanogaster* larvae were injected with 25 nM Cry1Aa, the wing disc died by necrosis (with cell swelling) and wingless adults emerged [32]. ABCC2s from other insect species also facilitate cell-swelling induction by Cry1A. *H. virescens* ABCC2 (HevABCC2) conferred susceptibility to Sf9 cells [33]. *Drosophila melanogaster* larvae expressing PxABCC2 in the midgut showed susceptibility and died upon feeding on a diet containing 0.05 ppm Cry1Ac [34]. As described above, the susceptibility-conferring ability of ABCB1 from *C. tremula* was confirmed using a heterologous expression system in Sf9 cells [17]. Therefore, heterologous expression of ABC transporters facilitates Cry1As-mediated induction of cell swelling. Cell swelling shown in these reports coincides well the symptom seen in the Cry1As intoxicated insect midgut [26]. 

### 2.2. BmABCC2 Facilitates Cation-Permeable Pore Formation by Cry1A Toxins

In artificial phospholipid membrane vesicles or planar lipid bilayers reconstituted with purified aminopeptidase N or GPI-linked receptor complex, Cry1A formed cation-permeable pores and transports Rb^+^ through the membrane [20,21,22]. An osmotic swelling assay showed that the internal diameter of the pore formed by Cry1Ac was approximately 2.4 nm [35]. In addition, apical-to-basal K^+^ flux was detected by short-circuit current measurement in the midgut of an insect treated with Cry1Aa and Ac [36]. Thus, Cry1A forms cations-permeable pores in lipid membranes. 

According to the Structural Classification of Proteins [37] (http://scop.mrc-lmb.cam.ac.uk/scop/), domain I of 3D-Cry toxins is the membrane translocation domain. The α-helices of this domain in colicin A and diphtheria toxin form pores or translocate to the cell membrane [38,39]. Furthermore, cell swelling is induced by water influx into the cytosol through the cell membrane [40], likely driven by the osmotic pressure generated by K^+^ influx [41]. Therefore, the facilitation of cation-permeable pore formation by ABCC2 suggests it is a functional receptor for 3D-Cry toxins. 

To determine whether it facilitates cation-permeable pore formation by 3D-Cry toxins, BmABCC2 was heterologously expressed in the *Xenopus* oocyte membrane and subjected to two-electrode voltage clamp assay. BmABCC2 facilitated the inward negative current (inward flow of cations) generated by Cry1Aa and Cry1Ab [42]. In addition, the magnitude of the inward negative current increased continuously; a similar result was obtained using the combination of BmABCC2 and Cry1Ab. Thus, it is likely that one molecule of BmABCC2 can continuously facilitate pore formation by Cry1A toxins, i.e., the cycle of Cry1A associating with BmABCC2, inserting α-helices into the membrane, and dissociating from BmABCC2 repeats continuously. Furthermore, after incubation with a high concentration of Cry1A, the pigment at the animal pole of *BmABCC2*-expressing *Xenopus* oocytes was dispersed and the cell membrane ruptured within 1 h [42]. The same principle may be in operation in the midgut columnar cells of insects exposed to Cry1A.

### 2.3. Binding to 3D-Cry Toxins

Receptors must be able to bind their target molecules. Thus, receptors in the midgut cell membrane were identified by screening for molecules that bind to 3D-Cry toxins using ligand blotting and pull-down assays. This resulted in the identification of cadherin-like receptor [43], aminopeptidase N [44], alkaline phosphatase [45], chlorophyllide-binding protein P252 [46,47], and BTR-270 glycoprotein [48] as candidate functional receptors. However, there was no report on the identification of ABCC2.

In a ligand blotting assay conducted in the authors’ laboratory, 10 ng BmABCC2 separated by sodium dodecyl sulfate polyacrylamide gel electrophoresis (SDS-PAGE) did not bind to Cry1Aa toxin, likely because of denaturation of its 12 membrane-spanning domains. Thus, the binding ability of ABCC2 might be lost upon disruption of its three-dimensional structure. To evaluate the binding ability of ABCC2, BmABCC2 was produced as a FLAG-tagged protein in Sf9 cells using a baculovirus expression system, solubilized with n-dodecyl-β-D-maltoside (used for X-ray crystallography of ABC transporters [49]), purified using an anti-FLAG^®^M2 affinity gel, and its binding ability was assessed by dot blotting and surface plasmon resonance (SPR) [50]. SPR showed that BmABCC2 had a very low dissociation rate and a dissociation constant (*K_D_*) of 3.1 × 10^−10^ M, indicating a high binding affinity for Cry1Aa [50].

The high binding affinity and promotion of cell-swelling induction and cations-permeable pore formation of Cry1A indicate that BmABCC2 is a functional receptor for Cry1A.

## 3. Role of ABC Transporters as Receptors for 3D-Cry Toxins

By gene silencing in larvae of *Spodoptera exigua* and *P. xylostella*, SeABCC3 and PxABCC3 were found to function as receptors for Cry1Ac [51,52]. In addition, heterologous expression of *SeABCC3* conferred susceptibility to 100 nM Cry1Aa and 1 µM Cry8Ca in HEK293T cells [31]. Similarly, *Spodoptera litura ABCC3* (*SlABCC3*) expression conferred susceptibility to 20 nM Cry1Ac in High Five cells [53]. Furthermore, BmABCC3 was linked to the induction of cell swelling by 100 nM Cry1Aa in HEK293T cells [54]. These suggests that ABCC3 also function as a receptor in lepidopteran insects. 

ABCC3 is most phylogenetically similar to ABCC2, and ABCC2 and ABCC3 looks to make lepidopteran-specific clade in a larger clade which include human ABCC4 (humABCC4) (humABCC4 clade, tentatively) [31,55] (Figure 1). This explains the specificity of Cry1A for lepidopteran insects [31]. In contrast, although Cry1Ca and Cry1Da are active in the larvae of *B. mori* and *S. exigua*, BmABCC2, BmABCC3, SeABCC2, and SeABCC3 did not function as receptors for these toxins in Sf9 or HEK293T cells [31,54] (Figure 1). Thus, Cry1Ca and Cry1Da looks not to target ABCC2 and ABCC3 at least in *B. mori* or *S. exigua*. Members of humABCC4 clade, ABCC1, ABCC4, ABCC5, ABCC6, ABCC7, and ABCC8 are close to ABCC2 and ABCC3 [55]. They could be candidate functional receptors for 3D-Cry toxins close to Cry1A (e.g., Cry1B, Cry1C, and Cry1D) (Figure 1). In contrast, *Tribolium castaneum* ABCC4 (TcABCC4), BmABCC2, and SeABCC3 functioned as receptors for Cry8Ca in HEK293T cells [31]. The clade formed by Cry8 and Cry9 is adjacent to the clade formed by Cry1 and Cry7 (Figure 1). TcABCC4 is assigned to the humABCC4 clade and is separated from BmABCC2/SeABCC3 by several factors [31,55] (Figure 1). Furthermore, several humABCC4-clade molecules (e.g., BmABCC4 and BmABCC6) are highly expressed in *B. mori* midgut cells [55]. Therefore, functional receptors for Cry7 and Cry9 that are active to *B. mori* are also likely present in the humABCC4 clade.

A laboratory-selected strain of the oligophagous leaf beetle *C. tremula* showed 6400-fold increased resistance to Cry3Aa in comparison to a susceptible strain; the increased resistance was related to one recessive allele [56]. A linkage analysis showed that deficiency in ABC transporter B1 (CtABCB1) mediated the increased resistance. Furthermore, in Sf9 cells, CtABCB1 functions as a receptor for Cry3Aa [17]. These data suggest that ABCB1 is the most important receptor for Cry3Aa.

Linkage mapping showed that resistance to Cry2Ab in *Helicoverpa armigera* and *Helicoverpa punctigera* was related to loss-of-function mutations of ABC transporter A2 (*ABCA2*) genes [15]. In addition, *HaABCA2*-knockout strains of *H. armigera* showed high-level resistance to Cry2Aa and Cry2Ab [16]. Knockout of BmABCA2 resulted in high-level resistance to Cry2Aa and Cry2Ab (Watanabe, personal communication). In addition, in our experiment, BmABCA2 facilitated induction of swelling by Cry2Aa and Cry2Ab in HEK293T cells (in preparation). Thus, ABCA2 must be considered to be the major functional receptor for Cry2Aa and Cry2Ab in these insects.

Cry1 and Cry2 are located at the most distant sides of the 3D-Cry toxin phylogenetic tree (Figure 1), whereas Cry3 is located in the middle. Therefore, 3D-Cry toxins may have adapted to use ABCA, ABCB, and ABCC during evolution. ABCB is closest to ABCC and ABCA is most distant from ABCC [57], and ABCD, ABCE, ABCF, ABCG, and ABCH are between ABCA and ABCC. Therefore, it is no wonder that functional receptors of the other 3D-Cry toxins will be found from these ABC transporters.

## 4. Receptor Functions of ABCC2, Cadherin-Like Receptor, and Aminopeptidase N

ABCC2 deficiency sometimes causes higher-level resistance than does deficiency in cadherin-like receptors [3,5,13]. To determine why, the receptor function of these two factors was investigated in non-susceptible cultured cells.

Sf9 cells expressing the toxin-binding region of BtR175, BtR175-TBR began to respond to Cry1Aa, Ab, and Ac at 100, 100, and 400 nM, respectively [30]. In contrast, BmABCC2-expressing Sf9 cells began to respond to Cry1Aa, Ab, and Ac at 100 pM, 10 nM, and 1 nM, respectively [30,50]. Thus, the susceptibility of BmABCC2-expressing cells is 1,000-, 10-, and 400-fold higher than that of cells expressing BtR175-TBR. In the other experiment, Sf9 cells expressing BtR175 were susceptible to 130 nM Cry1Aa [9]. Furthermore, HEK293T cells expressing BtR175-TBR variants were susceptible to 100 nM Cry1Aa [58]. In contrast, HEK293T cells expressing BmABCC2 began to respond (cell swelling) at 1 nM [54]. *Heliothis virescens* ABCC2 (HevABCC2) conferred high susceptibility to rupture by Cy1A on Sf9 cells [33]; however, the cadherin-like receptor of *H. virescens*, HevCaLP, conferred low susceptibility. In a *D. melanogaster* wing-disc expression system using Gal4-UAS, BmABCC2, but not BtR175-TBR, conferred susceptibility to 100 nM Cry1Aa [32]. BtR175 and BtR175-TBR have similar abilities to facilitate induction of cell swelling by 3D-Cry toxins [9,30]. Thus, ABCC2 looked to confer greater susceptibility to Cry1A than did cadherin-like receptor, irrespective of the heterologous expression system used. However, there is only the meager evidence of Western blotting for the equivalence of expression levels of ABCC2 and cadherin-like receptor. The cell swelling induction-facilitating activity of BmABCC2 and BtR175 cannot be compared based on the same number of expressing molecules. Therefore, it is difficult to say that differences in the receptor activity of these molecules are really due to the differences in the receptor function. 

The amount of complementary RNA (cRNA) of BmABCC2 and BtR175-BTR injected into the *Xenopus* oocyte expression system was adjusted to 500 ng [42]. In BmABCC2-expressing oocytes, the inward negative current increased within 100 s in the presence of 0.625 nM Cry1Aa [42], and 10 nM Cry1Ab also increased the inward negative current. However, in BtR175-TBR-expressing oocytes, no inward negative current was observed within 100 s, irrespective of the Cry1Aa concentration. Indeed, an inward negative current was detected only after 60 min. BmABCC2 looked to have 5,000-fold higher current induction-facilitating activity than BtR175-TBR when factors of time and Cry1Aa concentration was multiplied [42]. This suggests that BmABCC2 has greater Cry1A-receptor activity than does BtR175. This may explain why ABCC2 deficiency leads to greater resistance than does cadherin-like receptor deficiency.

Aminopeptidase N has been reported as an important functional receptor for 3D-Cry toxins, but its function is unclear. In an in vitro cell-burst assay using collagenase-dissociated *B. mori* midgut epithelial cells, anti-BtR175, but not anti-APN1 antiserum inhibited Cry1As [59]. Tanaka et al. investigated the receptor activity of *B. mori* APN1 (BmAPN1) using Baculovirus-Sf9 heterologous expression system and cRNA-injected *Xenopus* oocyte expression system [30,42]. Expression of BmAPN1 alone did not facilitate induction of cell swelling by Cry1Aa or a negative inward current. In addition, in the other reports, heterologously expressed APNs did not show receptor activity [60,61,62]. Although APN was reported in several other papers to facilitate toxin-induced cell swelling, the cells used in those papers appeared to have a problematic condition. Modification of APN by incomplete sugar chains may explain its lack of receptor activity in heterologous expression systems [63]. However, because *M. sexta* APN conferred susceptibility to Cry1Ac on *D. melanogaster* [64], it is unclear why Sf9 cells, lepidopteran insect cells cannot modify APN with sugar chains. Furthermore, Cry1Aa can bind to non-sugar-modified APN [65]. However, Cry1Aa did not use BmAPN1 as a functional receptor in heterologous expression systems [30,42]. Also, Cry1Ac formed K^+^ channels in artificial lipid membranes reconstituted with partially purified APN from *M. sexta* [20]. Cry1A promotes Rb+ release from phosphatidylcholine vesicles reconstituted with partially purified APNs from *H. virescens* and *M. sexta* [21,22]. However, APNs were contaminated with some proteins of 250–300 kDa, which is in the range of mass of putative ABCC2 dimer and P252 [24,46]. Thus, whether the electric current in, and Rb^+^ release from, the artificial membrane were dependent on APN is unclear. In addition, whether pore formation by 3D-Cry toxins detected in these experiments really means adequate activity in the induction of swelling and rupture of the real midgut columnar cells is unclear. APN reportedly facilitates pore formation after receiving oligomerized 3D-Cry toxins from cadherin-like receptor [25,66,67]. This may explain why APN did not facilitate induction of cell swelling when solely expressed in cultured cells. However, this contradicts the aforementioned hypothesis derived from the results of experiments using partially purified APN-containing artificial membranes [20,21,22], which lacked cadherin-like receptor.

No linkage-mapping study has shown that APN induces high-level resistance to 3D-Cry toxins, and thus the role of APN in the mode of action of 3D-Cry toxins remains obscure. This is reasonable if APN deficiency is lethal to the insect. In contrast, the susceptibility of insect larvae was reduced by downregulation of APN by RNA interference [68,69]. However, further work is required to confirm this, as the downregulation APN was inadequate and the decrease in susceptibility was obscure. Generating knockout insects by genome editing may help to clarify the role of APN as a receptor. Recently, APN was hypothesized to collect 3D-Cry protoxins and pass them to cadherin-like receptor [70].

## 5. Relationship between Binding Affinity to 3D-Cry Toxins and Receptor Activity 

The specificity and activity of 3D-Cry toxins should be determined by their receptor-binding affinity. The receptor activities of BmABCC2, BmABCC3, and BtR175 were evaluated using several heterologous expression systems, and their binding affinities to 3D-Cry toxins were determined by SPR [31,50,54]. The relationships between the 3D-Cry toxin-binding affinities and receptor activities of BmABCC2, BmABCC3, and BtR175 in cultured cells are shown in Table 1. In general, receptors with high toxin-binding affinities induced cell swelling in low concentration of 3D-Cry toxins. For example, only BmABCC2 and BmABCC2-R (BmABCC2 derived from a Cry1Ab resistant strain, Chinese No.2) with *K_D_* values <10^−10^ M responded to 0.1–10 nM 3D-Cry toxin (Table 1). In contrast, receptors with *K_D_* values >10^−7^ M did not facilitate induction of cell swelling by 1 µM Cry1Aa, Cry1Ab, Cry1Ca, Cry1Da, Cry3Bb, or Cry8Ca. For BmABCC3 and Cry1Aa, and TcABCC4A and Cry8Ca, a *K_D_* <10^−8^ M indicated receptor activity. However, some molecules with *K_D_* of 10^−8^ M did not function as receptors (Table 1). Furthermore, although the *K_D_* was 10^−10^ M, BmABCC2 showed low receptor activity for Cry1Ab (Table 1). This suggests that a factor other than binding affinity affects receptor function. The inability of BmABCC2-R, which was from a resistant *B. mori* strain, to function as a receptor for Cry1Ab despite a *K_D_* of 2.39 × 10^−8^ was considered to be due to inhibition of some processes after binding by insertion of a tyrosine at position 234 of BmABCC2 [30].

In contrast, although the *K_D_* for BtR175-TBR and Cry1Aa is 7.2 × 10^−10^ M [50], Sf9 cells expressing BtR175-TBR facilitated induction of cell swelling by >200 nM Cry1Aa [30] (Table 1). The molecular shape, distance between the membrane and the 3D-Cry toxin-binding site, and the angle of binding of 3D-Cry toxins might differ between BmABCC2 and BtR175-TBR. In addition, cadherin-like receptor induces oligomerization of 3D-Cry toxins [66,70]. Thus, to understand the reason of difference in the receptor activities of BmABCC2 and BtR175-TBR further studies are needed.

## 6. Structures of BmABCC2 Responsible for Its Function as a Cry1Aa Toxin Receptor

It is hypothesized that 3D-Cry toxins insert helices into the intramolecular pore of ABC transporters; indeed, this may explain the 1000-fold greater Cry1Aa-receptor activity of BmABCC2 than BtR175-TBR. Alternatively, the kinetic energy generated by opening the ABC transporter gate might trigger insertion of 3D-Cry toxins into the membrane. As receptors, Cry1A uses ABCC2 and ABCC3; Cry8Ca uses ABCC2, ABCC3, and ABCC4; Cry2A uses ABCA2; and Cry3Aa uses ABCB1 (Figure 1). Therefore, 3D-Cry toxins appear to have adapted to use different ABC transporters. The association of two nucleotide-binding domains using ATP molecules opens the intramolecular pore [18]. Quite unexpectedly, the receptor activity of two nucleotide-binding domain-deleted mutants of BmABCC2, D1del and D2del (Figure 2A), in the baculovirus-Sf9 heterologous expression system was similar to that of the wild type [24]. Thus, ABC transporters seem to have another reasons for playing the roles of a high functional receptor.

The extracellular loop (ECL) 1 and 4 domains of BmABCC2 were found to be related to its Cry1Aa receptor activity in deletion mutants generated using a baculovirus-Sf9 heterologous expression system (Figure 2), and the importance of ECL4 was confirmed using alanine-replacement mutants [24]. Recently, we further confirmed the importance of ECL1 using alanine-replacement mutants (in preparation). However, alanine replacement in ECL2, ECL3, ECL5, and ECL6 did not affect the receptor activity of BmABCC2 [24,54]. Thus, ECL1 and ECL4, the longest loops in BmABCC2, look important for its interaction with Cry1Aa (Figure 2).

In the High Five cell heterologous expression system, exchange of each ECL1–2 region between *S. litura* and *S. frugiperda* ABCC2 was indicated to be important to the specificity differences of Cry1Ac toxin [72], as was the amino acid at position 125 in ECL1. Also, Q^125^ of *H. armigera* ABCC2 was a key determinant of receptor activity [72].

The resistant *B. mori* race Chinese No. 2, which has an additional tyrosine residue at position 234 in ECL2 of BmABCC2 (Figure 2B), had 3000-fold lower susceptibility to Cry1Ab than the susceptible race Ringetsu [6]. Indeed, additive insertion of alanine, phenylalanine, arginine, tryptophan, and histidine at this site reduced the receptor activity of BmABCC2 from Ringetsu in a baculovirus-Sf9 heterologous expression system [71] (Figure 2B). In contrast, all of these BmABCC2 mutants retained a high level of receptor activity for Cry1Aa. Thus, ECL2 appears not to be important for direct binding to Cry1A, but may be related to specificity. In incompatible combinations (e.g., BmABCC2-R of Chinese No. 2 and Cry1Ab), ECL2 may hamper binding to, or the membrane insertion of, the Cry1Ab toxin [71].

ABCC2 and ABCC3 are closest in humABCC4 clade, but BmABCC2 conferred 1000-fold greater susceptibility to Cry1Aa on Sf9 cells than did BmABCC3 [30], and the *K_D_* of BmABCC2 for Cry1Aa was 100-fold lower than that of BmABCC3 [54] (Table 1). That is, the susceptibility-conferring activity of ABCC2 and ABCC3 is dependent on their binding affinity for Cry1Aa toxin. Several three serial amino acid residues (triplets) were swapped between BmABCC2 and BmABCC3, and two triplets from BmABCC2 ECL1 and ECL3, ^129^EAT^131^ and ^363^YIS^365^, respectively, increased the receptor activity of BmABCC3 [54] (Figure 2B). The ^129^EAT^131^ replacement reduced the *K_D_* (increased affinity) of BmABCC3 six-fold, indicating that ECL1 is an important determinant of the receptor activity of BmABCC2 and BmABCC3. In contrast, ^363^YIS^365^ replacement did not increase the binding affinity, suggesting that the increase in activity is caused by removal of structures that inhibit binding to 3D-Cry toxins or that hamper insertion of 3D-Cry toxins [54].

Comparisons of ABCC2 molecules from different insect species and of ABCC2 and ABCC3 from the same insect species have indicated the importance of ECL1 and ECL4 in determining receptor activity to Cry1A. ECL2 and ECL3 may also play roles in determining incompatibility. Therefore, ECL1, 2, 3, and 4 of ABCC2 and ABCC3 are determinants of the receptor activity and/or species specificity of Cry1A.

## 7. ABCC2-Binding Sites in Cry1Aa

The binding sites in Cry1Aa were analyzed by SPR using deletion mutants [50]. A loop-3 deletion mutant exhibited 1000-fold lower activity for facilitating induction of cell swelling in BmABCC2-expressing Sf9 cells and significantly lower toxin-binding affinity, suggesting that the BmABCC2-binding site is in loop 3. In contrast, a loop-2 deletion mutant had 1000-fold lower cell swelling induction-facilitating activity, but its toxin-binding affinity was unchanged. In addition, alanine replacement of three or four amino acids in the central region of loops 2 and 3 did not impact the binding affinity to BmABCC2, suggesting that the binding sites are located in the root of loops 2 and 3, which were not mutated. Mutants with unchanged binding affinity had long unchanged regions in the root of loops 2 and 3 [50]. The decreased binding affinity of the loop-3 deletion mutant may have been caused by structural distortion of the root region. In contrast, many mutants with a reduced ability to facilitate induction of cell swelling retained their Cry1Aa-binding affinity [50], suggesting that after binding to BmABCC2 the loops of Cry1Aa have some other roles in its insertion into the membrane. 

The cadherin-like receptor-binding sites of Cry1A toxins are in the loop region of domain II [73,74,75]. A binding inhibition assay showed that the BmABCC2- and BtR175-TBR-binding sites overlap in the loop region [50]. Thus, the binding pocket of Cry1Aa domain II consists of loops that bind not only to the structurally related BmABCC2 and BmABCC3 [31] (Table 1) but also to the structurally unrelated BtR175. That is, the binding pocket has a multiple binding property. Moreover, multiple binding property was also reported for domain III of Cry1a toxins [65]. Thus, Cry1A toxins has two multiple binding domains. This fact deeply impresses that *B. thuringiensis* is a pathogenic bacterium and uses toxins in hunting hosts. Although, several hypotheses have been proposed, the role of domain III in hunting is unclear. Only the loop region of domain II is adequately indicated to function as a binding site to functional receptors.

Binding of Cry1A to ABCC2 via ECL1 and ECL4 is unlikely to occur while the gate is open because ECL1 and ECL4 are separated [18]. In addition, binding of Cry1A to ABCC2 via the domain II loops would hamper insertion of the α-helices of the domain I into the intramolecular pore of ABCC2, because these loops seemed to cover the closed gate of the latent pore (Figure 2A). Thus, the α-helices likely access the lipid layer around ABCC2 molecules.

## 8. Synergism of ABCCs and Cadherin-Like Receptor and Roles in Susceptibility Determination

The receptor activity of BmABCC2 was 1000–5000-fold higher than that of BtR175-TBR [30,42], and so a deficiency in cadherin-like receptor should not cause resistance in principle. The susceptibility to Cry1Aa, Cry1Ab, and Cry1Ac of Sf9 cells expressing BtR175-TBR and BmABCC2 was 10-, 1000-, and 100-fold higher, respectively, than that of cells solely expressing BmABCC2 [30]. The significant increase in susceptibility caused by double expression was due to synergism between the two molecules [30]; indeed, HevCaLP and HevABCC2 also have synergistic activity [33]. Such synergism seems to be a major determinant of susceptibility to 3D-Cry toxins, because deficiency of cadherin-like receptor or ABCC2 induces resistance in the same insect species. Although the activity of a Cry1Ac mutant to cadherin-like receptor-knockout or ABCC2-knockout *Trichoplusia ni* strains decreased only slightly, it was decreased 3700-fold by knockout of both. Thus, the synergism of ABCC2 and cadherin is a determinant of the susceptibility of *T. ni* larvae [76].

Formation of the cation-permeable pore in the cell membrane of *Xenopus* oocytes was accelerated eight-fold by co-expression of BtR175-TBR and BmABCC2 compared to expression of BmABCC2 only. Moreover, expression of BtR175-TBR alone did not lead to formation of cations-permeable pores. In contrast, the amount of membrane-bound oligomer was similar in BtR175-TBR- and BmABCC2-expressing oocytes [42]. Therefore, oligomers from BtR175-TBR-expressing oocytes should have remained on the membrane surface. In contrast, most oligomers from BmABCC2-expressing oocytes were considered to be inserted into the membrane. Because cadherin-like receptor has high 3D-Cry toxin oligomerization activity [25,67], the synergistic effect of BtR175-TBR and BmABCC2 is likely dependent on oligomer supply by BtR175 and promotion of pore formation by BmABC2. *Plutella xylostella* ABCC2 facilitates membrane insertion of Cry1Ac oligomers prepared by incubation with the *M. sexta* cadherin fragment CR7-CR12 [77]. It is uncertain whether 3D-Cry toxin is passed from cadherin-like receptor to ABCC2 as an oligomer or monomer, or whether ABCC2 facilitates membrane insertion of 3D-Cry toxin as a monomer or oligomer. Because the BmABCC2-binding site on Cry1Aa overlaps that of BtR175-TBR [50], BmABCC2 and BtR175-TBR should inhibit each other’s binding to Cry1Aa. Therefore, a single Cry1Aa molecule is unlikely to be translocated from BtR175-TBR to BmABCC2. If this is the case, pore formation may involve aggregation of several Cry1Aa toxin monomer-holding BtR175-TBRs and monomer-holding BmABCC2s, followed by oligomerization of Cry1Aa and its membrane insertion with the assistance of BmABCC2. The putative roles of ABCC2, ABCC3, and cadherin-like receptor (based on the roles of BmABCC2, BmABCC3, and BtR175 in *B. mori* larvae) are shown in Figure 3. A signal transduction domain-deleted BtR175 mutant, BtR175-DEL, had the same ability to facilitate the induction of cell swelling as BtR175-TBR. In addition, the combination of BtR175-DEL and BmABCC2 exerted a synergistic effect in inducing cell swelling, similar to the combination of BtR175-TBR and BmABCC2 [78]. Therefore, oncosis-like programmed cell death induced by the signal-transduction domain of cadherin-like receptor is not relevant to the facilitation of induction of cell swelling by BtR175-TBR or the synergistic effect of BmABCC2 plus BtR175-TBR or that of BmABCC3 plus BtR175-TBR. 

A recombinant *M. sexta* cadherin essential toxin-binding region protein (CR12-MPED) enhanced the activity of Cry1Ac in *M. sexta*, *H. virescens*, and *Heliothis zea* larvae [79], in agreement with several prior reports [51,80,81]. This enhancement and synergistic effect may have the same underlying mechanism. In contrast, in the other reports, the toxin-binding region of cadherin did not increase the activity of Cry1 toxins [75,82]. In addition, in our experiment, recombinant BtR175-TBR did not enhance the activity of Cry1Aa in the BmABCC2 expressing Sf9 cells. Further studies of the relationship between enhancement of the activity by the toxin-binding region of cadherin and the synergistic effect of BtR175-TBR and BmABCC2 are thus warranted.

Cry1Aa, Cry1Ab, and Cry1Ac induced swelling of BmABCC2-expressing Sf9 cells at 100 pM, 10 nM, and 1 nM, respectively [30]. In contrast, the LC_50_ of Cry1Aa, Cry 1Ab, and Cry1Ac in *B. mori* larvae was 0.45, 0.26, and 1.5 µg/g, respectively [30]. Thus, the larvicidal activity of Cry1A toxins is not correlated with their induction of cell swelling in the BmABCC2-expressing Sf9 cells. This suggests that in *B. mori* larvae, several factors determine susceptibility to Cry1A toxins, including BtR175-TBR and BmABCC2. In addition, in our experiment, BtR175-TBR and BmABCC3 also exerted a synergistic effect in HEK293T cells. Thus, in *B. mori* larvae, not only BmABCC2 but also BmABCC3 may be involved in determining susceptibility to Cry1Aa through synergism with BtR175 (Figure 3). This is in agreement with the finding that knockout of BtR175, but not BmABCC2, induces resistance to Cry1Aa in *B. mori* larvae (Watanabe, personal communication). In contrast, mutation of BmABCC2 alone increased resistance to Cry1Ab 300-fold in *B. mori* larvae [6]. Also, susceptibility to Cry1Ab was rescued by introduction of *BmABCC2* cDNA from a susceptible to a resistant *B. mori* strain [6]. Moreover, HEK293T cells expressing BmABCC3 were not susceptible to Cry1Ab [54]. Therefore, in *B. mori* larvae, BmABCC3 is unlikely to be a determinant of susceptibility to Cry1Ab. Similarly, ABCC3 seems not to be related to the susceptibility of insects, since 2800-fold increased resistance to Cry1Ac in *P. xylostella* and 1000-fold increased resistance to Cry1Ab and Cry1Ac in *H. armigera* [5,13] were generated by ABCC2 deficiency alone. Thus, roles of ABCC2, ABCC3, and cadherin-like receptor (Figure 3) in the determining susceptibility seems to be different in each insect even to Cry1A toxins.

## 9. Production of Multiple Toxins by *B. Thuringiensis*

Several strains of *B. thuringiensis* produce multiple toxins, including 3D-Cry toxins, e.g., Cry1A, Cry1B, Cry1C, Cry1D and Cry2 [83,84]. A *P. xylostella* strain, NO-QA was resistant to Cry1A toxins but not to Cry1B, Cry1C, or Cry1D [85]. Conversely, *P. xylostella* BCS-Cry1C-2 was resistant to Cry1C but susceptible to Cry1A [86]. These suggests that several toxins use their original factors as receptors. Furthermore, an ABCA2 truncation-dependent Cry2Ab-resistant *H. armigera* strain was susceptible to Cry1Ac [15]. In addition, Cry1Aa, but not Cry1C or Cry1D, use ABCC2 and ABCC3 [54] (Figure 1). Therefore, simultaneous production of multiple 3D-Cry toxins that use different receptors promotes survival in *B. thuringiensis* by preventing development of host resistance. The *B. thuringiensis* subspecies *israelensis* (Bti) is used to control mosquitoes and black flies; however, no outbreak of highly resistant strains has been reported [87,88]. This might be due to production of Cry4Aa, Cry4Ba, Cry10Aa, Cry11Aa, Cyt1Aa, and Cyt2Ba by Bti. Alternatively, simultaneous production of multiple toxins may function simply to increase insecticidal activity, e.g., simultaneous production of Cry1Ca and Cry9Aa exerted a synergistic effect against *H. armigera* [89]. In addition, synergism between 3D-Cry and Cyt toxins and between 3D-Cry toxins and vegetative insecticidal protein has been reported [90,91]. The utility of this strategy used by *B. thuringiensis* is increasingly useful to Bt spray and 3D-Cry toxin gene-integrated GMOs.

## Figures and Tables

**Figure 1 toxins-11-00124-f001:**
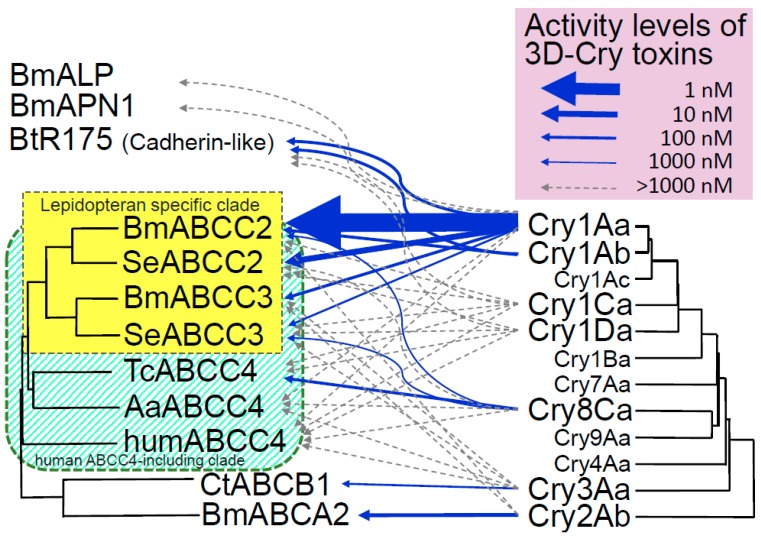
Use of ABC transporters as functional receptors by 3D-Cry toxins. Receptor activity data for Cry2Ab using ABC transporter-expressing HEK293T cells or Sf9 cells and previous results for Cry toxins [17,30,31,54]; arrow width indicates the level of activity. Phylogenetic tree of ABC transporters generated using the amino acid sequences of BmABCC2 (BAK82127.1), SeABCC2 (AIB06821.1), BmABCC3 (XP_012547933.1), SeABCC3 (AIB06823.1), TcABCC4 (XP_969849.1), AaABCC4 (APG42670.1), humABCC4 (NP_005836.2), CtABCB1 (APK18402.1), and BmABCA2 (ALE60402.1) with CLUSTAL W. Bm, *Bombyx mori*; Ct, *Chrysomela tremula*; Se, *Spodoptera exigua*; Tc, *Tribolium castaneum*; Aa, *Aedes albopictus*. A part of the phylogenetic tree of Cry toxins (*Bacillus thuringiensis* Toxin Nomenclature; http://www.lifesci.sussex.ac.uk/home/Neil_Crickmore/Bt/intro.html).

**Figure 2 toxins-11-00124-f002:**
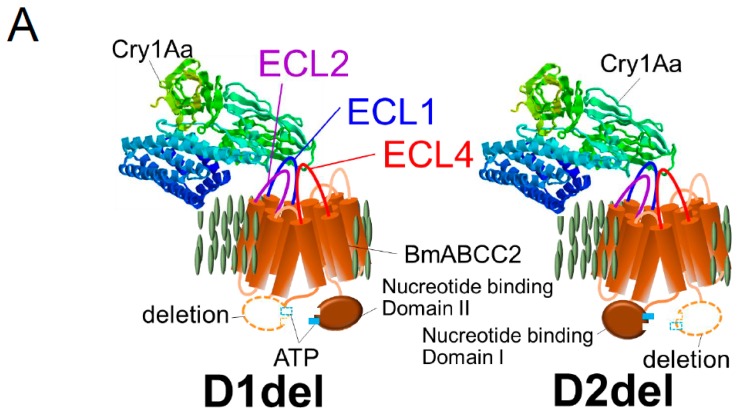
BmABCC2 structures that affect Cry1As receptor activity. (**A**) Nucleotide-binding domain-deleted BmABCC2 mutants, D1del and D2del, that exhibit receptor activity for Cry1Aa when expressed in the membranes of Sf9 or HEK293T cells [24]. ECL, extracellular loop. (**B**) Mutation sites in BmABCC2 that affect receptor activity for Cry1Aa or Cry1Ab [24] and a site at which amino acid replacement increases BmABCC3 receptor activity [54].

**Figure 3 toxins-11-00124-f003:**
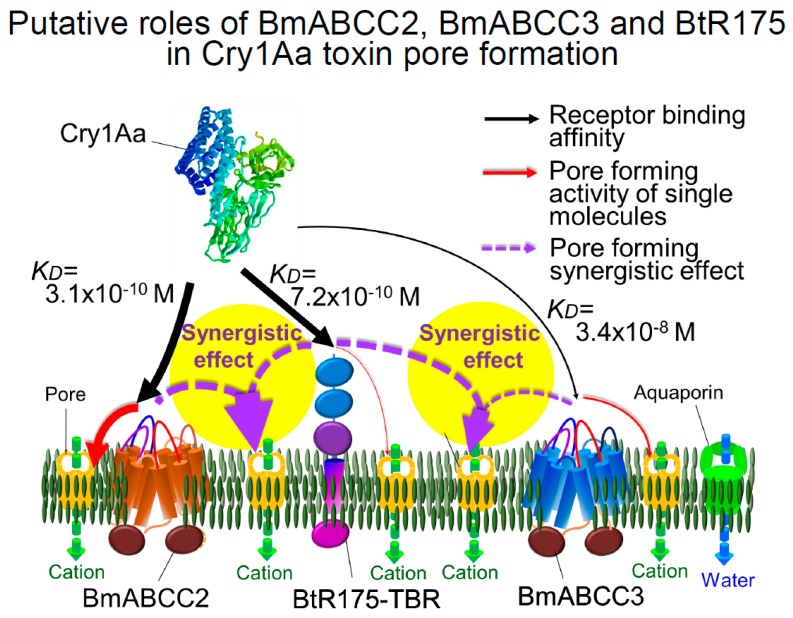
Putative roles of BmABCC2, BmABCC3, and BtR175 in Cry1Aa pore formation. Arrow thickness represents Cry1Aa receptor interaction and pore formation activities [30,31,42,50].

**Table 1 toxins-11-00124-t001:** Partial correlation between binding affinity and cell swelling activity of Cry toxins.

Receptor.	Toxin	*K_D_* (M) ^1^	Effective Conc. (nM) in Cell Swelling Assay ^2^
BmABCC2	Cry1Aa	3.1 × 10^−10^	0.1
	Cry1Ab	2.6 × 10^−10^	100
	Cry1Ca	1.7 × 10^−7^	>10,000
	Cry1Da	2.3 × 10^−6^	>2000
	Cry3Bb	2.0 × 10^−5^	>500
	Cry8Ca	1.9 × 10^−5^	1000
BmABCC2 from resistant line	Cry1Aa	2.8 × 10^−10^	10
	Cry1Ab	2.4 × 10^−8^	>1000
BmABCC3	Cry1Aa	3.4 × 10^−8^	100
	Cry1Ab	6.9 × 10^−8^	>4500
	Cry1Ca	3.9 × 10^−7^	>1000
	Cry1Da	4.2 × 10^−4^	>1000
	Cry3Bb	4.0 × 10^−8^	>1000
TcABCC4A	Cry1Aa	not detectable	>1000
	Cry1Ca	not detectable	>2500
	Cry1Da	not detectable	>2000
	Cry3Bb	1.8 × 10^−5^	>1700
	Cry8Ca	4.0 × 10^−8^	100
BtR175(BmCad)	Cry1Aa	7.2 × 10^−10^	200

^1^ The values of *K_D_* were sited from reports of Adegawa et al. [50] and Endo et al. [31,54]. ^2^ Lowest concentrations of toxins by which swollen cells were obviously induced were sited from reports of Tanaka et al. [30,71] Adegawa et al. [50] and Endo et al. [31,54].

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
