# Peer review of "Function and Role of ATP-Binding Cassette Transporters as Receptors for 3D-Cry Toxins"

_toxins, 2019, doi:10.3390/toxins11020124_

Reviewer 1 Report

            In this review, the authors explained the experimental results that indicate to ABC transporter act as the receptor for 3D-Cry toxin family. The authors report on the activity of cell swelling triggered by 3D-Cry toxin with protein receptors. In addition, the protein receptors are also relate to the process of insect larvae resistance against Cry toxins. The authors comment on the function of ABCC2 transporter and compare the activity to other receptors function. The authors also provided information of ABC transporter-Cry toxin interaction domains. The summary of ABC transporter as a receptor for Cry toxin will let readers to obtain the information of new possibilities of the Cry toxin mechanism and also will associate the understanding of insect resistance to bioinsecticide, Cry toxins.

However, this review needs English editing before publication. The manuscript also needs to be improved (minor revisions).

The following comments could help the authors improving the review:

1. The authors used term of 3D-Cry toxin. The readers will understand better if the authors give more information of classification of Cry toxin and their molecular structure.

2. In the title, the authors used the term of ATP-binding cassette transporters but in the main text used nucleotide-binding domain. The authors should be consistent.

3. The authors should provide the general information of ATP-binding cassette transporters e.g. its mechanism (before and after binding of ATP).

4. Line 51-54, the term of factor and determinant refer to protein receptors? It should be indicated.

5. Line 105-107, the authors usually talk about cell swelling of both cell types (later also indicated as necrosis). It  would be good to remove this sentence to avoid confusion.

6. Line 115, ABCB1, do the authors mean ABCB1 gene or ABCB1 protein?

7. Line 125, the authors should write Cry1AC instead of only AC and somewhere else below this section (line 208-209).

8. Please indicate the reference of the sentence in Line 291-293.

9. Line 338-340, the meaning is not clear. Please, rewrite the sentence.

10. Summary of the Figure 2 and 3 with tables would help the audience to clearly understand each domain of BmABCC2 and protein receptors.

11. At the final section, the authors should add a table containing each protein receptor, BmABCC2 and 3D-Cry toxins.

Author Response

Answers to the comments #1

In this review, the authors explained the experimental results that indicate to ABC transporter act as the receptor for 3D-Cry toxin family. The authors report on the activity of cell swelling triggered by 3D-Cry toxin with protein receptors. In addition, the protein receptors are also relate to the process of insect larvae resistance against Cry toxins. The authors comment on the function of ABCC2 transporter and compare the activity to other receptors function. The authors also provided information of ABC transporter-Cry toxin interaction domains. The summary of ABC transporter as a receptor for Cry toxin will let readers to obtain the information of new possibilities of the Cry toxin mechanism and also will associate the understanding of insect resistance to bioinsecticide, Cry toxins.

However, this review needs English editing before publication. The manuscript also needs to be improved (minor revisions).

The following comments could help the authors improving the review:

1. The authors used term of 3D-Cry toxin. The readers will understand better if the authors give more information of classification of Cry toxin and their molecular structure.

Response 1: According to the reviewer’s suggestion, 3D-Cry was explained as fallow. “Bacillus thuringiensis is the most widely used bio-pesticide due to several insecticidal proteins including Cry toxins. The largest group of Cry toxins is called three-domain Cry (3D-Cry) toxin, since its activated form is composed of three domains. 3D-Cry toxin genes are essential for the generation of genetically modified insect resistant crops.”

2. In the title, the authors used the term of ATP-binding cassette transporters but in the main text used nucleotide-binding domain. The authors should be consistent.

Response 2: We understand what the reviewer want to suggest. However, “ATP-binding cassette” is used to call the molecular name of the transporter and “nucleotide-binding domain” is used to call the domain name. Many specialists of ABC transporter differentially uses two terms like this.  

3. The authors should provide the general information of ATP-binding cassette transporters e.g. its c (before and after binding of ATP).

Response 3: As the reviewer suggested, explanation of gate-opening mechanism of ATP-binding cassette transporters is important. However, a short explanation “The association of two nucleotide-binding domains using ATP molecules opens the intramolecular pore [18]” exists in page 14.

4. Line 51-54, the term of factor and determinant refer to protein receptors? It should be indicated.

Response 4: According to the suggestion, “factor” was replaced with “receptor”.

5. Line 105-107, the authors usually talk about cell swelling of both cell types (later also indicated as necrosis). It  would be good to remove this sentence to avoid confusion.

Response 5: As the reviewer suggested, this sentence was difficult to understand. To avoid confusion we added explanation.

6. Line 115, ABCB1, do the authors mean ABCB1 gene or ABCB1 protein?

Response 6: According to the suggestion, ABCB1 was rewritten as a protein in a manner consistent with the other parts.

7. Line 125, the authors should write Cry1AC instead of only AC and somewhere else below this section (line 208-209).

Response 7: According to the suggestion, Ab and Ac were replaced with Cry1Ab and Cry1Ac.

8. Please indicate the reference of the sentence in Line 291-293.

Response 8: This place is only a speculation. “might” was added.

9. Line 338-340, the meaning is not clear. Please, rewrite the sentence.

Response 9: According to the suggestion, the sentence was rewritten.

10. Summary of the Figure 2 and 3 with tables would help the audience to clearly understand each domain of BmABCC2 and protein receptors.

Response 10: We understand what the reviewer want to suggest. However, Figure 2 and 3 themselves are summarizing the concepts of sections 6, 7, and 8 to help the audience.

11. At the final section, the authors should add a table containing each protein receptor, BmABCC2 and 3D-Cry toxins.

Response 11: I am sorry, but I cannot understand the meaning of this suggestion.

Reviewer 2 Report

Dear Author,

The current review article entitled " Function and role of ATP-binding cassette transporters as receptors for 3D-Cry toxins" focused on the biochemical, cell biological, and physical studies of the role of ABC transporters 87 as receptors for 3D-Cry toxins.

Line no: 181-185: Author may explain why this much resistance happens. What type of modification takes place in the recessive allele.

A conclusion section must be needed because the article is very comprehensive and to increase the readability it is important.

Figure 1: clade 3 is not specified like above two.

Author Response

Answers to the comments #2

The current review article entitled " Function and role of ATP-binding cassette transporters as receptors for 3D-Cry toxins" focused on the biochemical, cell biological, and physical studies of the role of ABC transporters 87 as receptors for 3D-Cry toxins.

Line no: 181-185: Author may explain why this much resistance happens. What type of modification takes place in the recessive allele.

I think this is outside the scope of this review and the original paper [56] is proposing a hypothesis.

A conclusion section must be needed because the article is very comprehensive and to increase the readability it is important.

I did not add a conclusion section because of the status of this manuscript (galley proof) and existence of an abstract which plays the same role as conclusion section.

Figure 1: clade 3 is not specified like above two.

I could not understand the suggestion.